# Neural Encoding and Decoding at Scale

**Yizi Zhang**[*1,4]   **Yanchen Wang**[*1]   **Mehdi Azabou**[1]   **Alexandre Andre**[2]   **Zixuan Wang**[1]   **Hanrui Lyu**[3]
**The International Brain Laboratory**[4]   **Eva Dyer**[2]   **Liam Paninski**[1,4]   **Cole Hurwitz**[1,4]

[1]Columbia University   [2]University of Pennsylvania   [3]Northwestern University
[4]The International Brain Laboratory   [*]Equal Contribution

## Abstract

Recent work has demonstrated that large-scale, multi-animal models are powerful tools for characterizing the relationship between neural activity and behavior. Current large-scale approaches, however, focus exclusively on either predicting neural activity from behavior (*encoding*) or predicting behavior from neural activity (*decoding*), limiting their ability to capture the bidirectional relationship between neural activity and behavior. To bridge this gap, we introduce a multimodal, multi-task model that enables simultaneous **N**eural **E**ncoding and **D**ecoding at **S**cale (**NEDS**). Central to our approach is a novel multi-task-masking strategy, which alternates between neural, behavioral, within-modality, and cross-modality masking. We pretrain our method on the International Brain Laboratory (IBL) repeated site dataset, which includes recordings from 83 animals performing the same visual decision-making task. In comparison to other large-scale models, we demonstrate that NEDS achieves state-of-the-art performance for both encoding and decoding when pretrained on multi-animal data and then fine-tuned on new animals. Surprisingly, NEDS's learned embeddings exhibit emergent properties: even without explicit training, they are highly predictive of the brain regions in each recording. Altogether, our approach is a step towards a foundation model of the brain that enables seamless translation between neural activity and behavior. Project page and code: https://ibl-neds.github.io/

. Correspondence to: Yizi Zhang <yz4123@columbia.edu>.

*Proceedings of the 42^{nd} International Conference on Machine Learning*, Vancouver, Canada. PMLR 267, 2025. Copyright 2025 by the author(s).

## 1. Introduction

Systems neuroscience has experienced a revolution in data acquisition, from brain-wide recordings using Neuropixels probes (Jun et al., 2017; Steinmetz et al., 2019; 2021; IBL et al., 2023; Khilkevich et al., 2024; Chen et al., 2024), to automated tracking of behavior from video data (Mathis et al., 2018; Pereira et al., 2022; Biderman et al., 2024). While these multi-animal, multimodal datasets open new avenues for understanding the relationship between neural activity and behavior, they also necessitate the development of analysis approaches that can effectively leverage massive volumes of data (Paninski & Cunningham, 2018; Stringer & Pachitariu, 2024).

Recent efforts to address this challenge have focused on developing large-scale models that can be trained across multiple sessions and animals (Azabou et al., 2024; Ye et al., 2024; Zhang et al., 2024b;a; Posani et al., 2024; Wang et al., 2025). These works demonstrate that with increasing scale of neural data comes improved performance on downstream tasks such as predicting neural activity from behavior (*encoding*) or predicting behavior from neural activity (*decoding*). Despite their promise, most preexisting large-scale modeling approaches are constrained by their task-specific nature, focusing exclusively on either encoding (Posani et al., 2024; Wang et al., 2025) or decoding (Azabou et al., 2024; Ye et al., 2024; Zhang et al., 2024a). This limits their ability to model the bidirectional relationship between neural activity and behavior. Overcoming this limitation will require multimodal models capable of seamlessly translating between different modalities.

To address this need, we propose a method for simultaneous **N**eural **E**ncoding and **D**ecoding at **S**cale (**NEDS**). Our approach leverages recent advances in self-supervised learning and multimodal masked modeling (He et al., 2022; Mizrahi et al., 2023), training a transformer-based model to jointly capture the relationship between neural activity and behavior. We utilize a multi-task-masking strategy (Tay et al., 2022; Zhang et al., 2024b) that alternates between neu-

ral, behavioral, within-modality, and cross-modal masking (shown in Figure 1A). After training, NEDS can perform decoding by masking behavior and predicting it from neural activity, or encoding by masking neural activity and predicting it from behavior. Our multi-task-masking framework unifies encoding and decoding, enabling flexible and scalable analysis of neural activity and behavior.

We evaluate our approach on the International Brain Laboratory (IBL) repeated site dataset (IBL et al., 2022), which consists of Neuropixels recordings targeting the same brain regions across 83 mice performing the same decision-making task. We benchmark NEDS on encoding and decoding of key task variables including whisker motion, wheel velocity, choice, and the "block" prior (Findling et al., 2023). We first demonstrate that NEDS outperforms an equivalent unimodal encoding and decoding method. We then compare NEDS to preexisting large-scale modeling approaches including POYO+ (Azabou et al., 2025) and NDT2 (Ye et al., 2024). We demonstrate that NEDS achieves superior performance compared to these approaches when pretrained on trial-aligned data from 73 animals and fine-tuned on data from 10 held-out animals. Finally, we show that pretrained NEDS exhibits emergent properties; without explicit training, the latent representations learned by NEDS are highly predictive of the brain regions in each session. Taken together, NEDS represents a new paradigm for large-scale neurobehavioral modeling, bringing us closer to a foundation model of the brain at single-cell, single-spike resolution. The contributions of this work include:

- A unified modeling approach for encoding and decoding of behavior at scale (NEDS) that achieves state-of-the-art performance in both tasks.

- A demonstration that both encoding and decoding performance scale meaningfully with the amount of pretraining data and model capacity (i.e., scaling).

- A demonstration of emergent properties: pretraining NEDS across many animals produces embeddings that accurately predict the brain regions in each session.

## 2. Related Work

***Neural encoding and decoding models.*** Neural decoding and encoding are complimentary analyses in systems neuroscience. Decoding quantifies *what* information is present in the neural activity while encoding quantifies *how* neural activity represents this information (Paninski & Cunningham, 2018). Traditional decoding models include linear regression, reduced rank regression, or more recently, deep learning approaches (Glaser et al., 2020). Improvements in decoding methodologies have significant potential for real-world applications, such as brain–computer interfaces, which have seen remarkable progress over the past

decade (Gilja et al., 2015; Pandarinath et al., 2017; Willett et al., 2023; Metzger et al., 2023). Encoding models classically take the form of linear or "generalized linear models" (GLMs) (Paninski, 2004; Truccolo et al., 2005). Again, recent advancements in deep learning have led to considerable improvements in this domain (Wen et al., 2018; Schrimpf et al., 2018; Ustyuzhaninov et al., 2022; Willeke et al., 2022; Li et al., 2023; Turishcheva et al., 2024; Wang et al., 2025). Encoding models can provide insights into the brain, such as how uninstructed movements dominate neural variability (Musall et al., 2019) or how neural correlates of decision-making are distributed across many brain regions (IBL et al., 2023).

***Multimodal models of neural activity and behavior.*** Recent advancements in large-scale electrophysiology and video tracking have facilitated the collection of massive neurobehavioral datasets. This has led to the development of novel multimodal approaches for understanding the relationship between neural activity and behavior (Sani et al., 2021; Hurwitz et al., 2021b; Gondur et al., 2023; Sani et al., 2024; Schulz et al., 2025). One common strategy is to use latent variable modeling to learn a shared embedding space for neural and behavioral data (Hurwitz et al., 2021a), as demonstrated by methods like PSID (Sani et al., 2021) and TNDM (Hurwitz et al., 2021b). More recently, masked modeling approaches, such as the masked VAE (M-VAE) proposed in a preprint by Schulz et al. (2025), have shown promise. In this work, structured masking is used to learn the conditional distributions between neural activity and behavior. Similar to NEDS, this allows for performing encoding and decoding after training. A limitation of M-VAE is its sequential VAE architecture (RNN-VAE), which poses scaling challenges and limits its applicability to multi-animal datasets.

***Large-scale models for neural analysis.*** Traditional analyses of neural data and behavior have been limited to single-animal, single-session models. Recent evidence suggests that this paradigm neglects shared information between animals performing similar tasks (Safaie et al., 2023; Zhang et al., 2024a). To exploit this shared information, ongoing work has focused on developing large-scale models that can be trained on neural data and behavior from many animals and brain regions (Pandarinath et al., 2018; Lurz et al., 2020; Azabou et al., 2024; Ye et al., 2024; Zhang et al., 2024a;b; Vermani et al., 2024; Wang et al., 2025). These works draw inspiration from the success of large-scale foundation models in natural language processing (Radford et al., 2018; 2019; Achiam et al., 2023) and the natural sciences (Jumper et al., 2021; Abramson et al., 2024), which demonstrate remarkable generalization abilities driven by broad pretraining. Notable examples of large-scale models for neural analysis include POYO+ (Azabou et al., 2025), a multi-task, multi-animal decoding approach based on POYO (Azabou et al., 2024) and NDT2, a masked modeling approach for

neural prediction that can be fine-tuned for decoding (Ye et al., 2024). Critically, both these models are task-specific: POYO+ is restricted to decoding and NDT2 can be fine-tuned for decoding but not encoding. To improve the scientific utility of large-scale neural models, we argue that they must be capable of performing both encoding and decoding.

## 3. Methods

In this section, we present a multimodal, multi-task model for **N**eural **E**ncoding and **D**ecoding at **S**cale (**NEDS**). The key innovation of our approach lies in the use of multi-task-masking in which NEDS alternates between neural, behavioral, within-modality, and cross-modal masking during training (Figure 1A). This approach enables NEDS to learn the conditional expectations between neural activity and behavior, unifying neural encoding and decoding within a single framework. We implement NEDS as a multimodal transformer in which each modality is tokenized independently and processed through a shared transformer (Figure 1B). NEDS enables scalable and flexible analysis of neural activity and behavior from massive, multi-animal datasets.

### 3.1. Dataset

For all analyses in this paper, we use the IBL repeated site dataset (IBL et al., 2022). This dataset consists of Neuropixels recordings collected from 10 labs with standardized experimental pipelines. The recordings target the same five brain regions across 83 adult mice performing a complex decision-making task. We train on data from up to 73 animals, holding out 10 animals for evaluation. One animal in the training set has two insertions, resulting in a total of 74 training sessions. For the neural data, we use trial-aligned, spike-sorted data (Chapuis et al., 2022), excluding neurons with firing rates less than 2 Hz. We utilize a total of 27,380 neurons from 225 brain regions We bin the neural activity using 20ms windows and fix the trial-length to 2 seconds (100 time bins). For the behavioral data, we use four task variables: whisker motion, wheel speed, choice (left/right), and the "block" prior (Findling et al., 2023). The block is the prior probability of the stimulus appearing on the left or right side: (1) 20/80% right, (2) 80/20% left, or (3) 50/50%. Whisker motion and wheel velocity are binned at 20ms resolution. We exclude trials in which mice exhibited reaction time outliers, as defined by the IBL Brain-Wide Map (IBL et al., 2023). We also evaluate our approach on a monkey reaching dataset (Pei et al., 2021) detailed in Appendix H.

### 3.2. Multi-task-masking for neural activity and behavior

Previous work has demonstrated that multi-task-masking, which involves alternating between different masking schemes during pretraining, is a powerful approach for de-

veloping large-scale generalist models (Tay et al., 2022; Zhang et al., 2024b). We extend this paradigm to model the relationship between neural activity and behavior. Concretely, let us denote $X$ as neural activity and $Y$ as set of behaviors the animal is performing. We define a set of masking schemes as follows (shown in Figure 1A):

- **Neural masking.** The neural activity $X$ is fully masked, and the model learns to predict it based on behavior $Y$. This masking scheme trains the model to predict the conditional expectation $\mathbb{E}[X \mid Y]$, effectively teaching it to perform neural encoding. Importantly, we alternate between predicting neural activity using *all* behavioral variables and using *one* behavioral variable at a time. This approach allows the model to perform encoding with respect to each behavioral variable during inference, enabling the ranking of each behavior's importance in explaining neural variability.

- **Behavior masking.** The behavior $Y$ is fully masked, and the model learns to predict it based on neural activity $X$. This masking scheme trains the model to predict the conditional expectation $\mathbb{E}[Y \mid X]$, effectively teaching it to perform neural decoding.

- **Within-modality random masking.** Random tokens are masked and then reconstructed from either the neural activity $X$ or behavior $Y$. This masking scheme, which models the conditional expectations $\mathbb{E}[X_{\text{masked}} \mid X_{\text{unmasked}}]$ and $\mathbb{E}[Y_{\text{masked}} \mid Y_{\text{unmasked}}]$, improves the model's ability to learn modality-specific representations by capturing intra-modal dependencies.

- **Cross-modal random masking.** Random tokens are masked and then reconstructed from both the neural activity $X$ and behavior $Y$. This masking scheme, which models the conditional expectation $\mathbb{E}[X_{\text{masked}}, Y_{\text{masked}} \mid X_{\text{unmasked}}, Y_{\text{unmasked}}]$, improves cross-modal learning by encouraging the model to learn the joint relationship between the modalities.

Each of these masking schemes encourages the model to capture a different aspect of the relationship between neural activity and behavior. By mixing between these masking schemes during training, NEDS is be able to seamlessly switch between encoding and decoding at inference, achieving state-of-the-art performance in both tasks. We ablate these masking schemes in Appendix B and find that using all masking schemes together yields the best performance.

### 3.3. Architecture

***Modality-specific tokenization.*** We map both neural and behavioral data to token sequences using modality-specific tokenizers (i.e., linear embeddings). For the neural data,

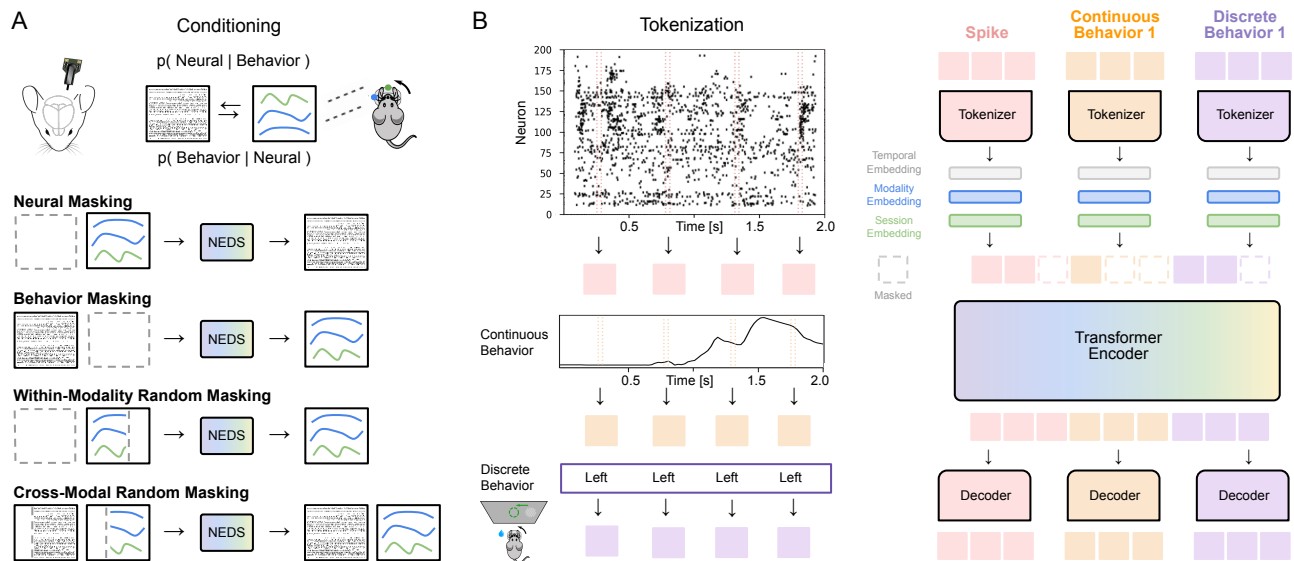

*Figure 1. Schematic illustration of NEDS .* (A) Neural encoding and decoding can be interpreted as modeling the conditional probability distributions between neural activity and behavior (Schulz et al., 2025). In NEDS, we utilize a multi-task-masking approach (Tay et al., 2022; Zhang et al., 2024a) to model the conditional expectations of these distributions as well as to encourage cross-modal and within-modality representation learning. This is achieved by alternating between neural, behavioral, within-modality, and cross-modal masking during training. (B) We implement NEDS using a multimodal transformer-based architecture. We utilize modality-specific tokenizers that convert spike counts and continuous behaviors into 20ms temporal tokens and discrete behaviors into sequences of repeated tokens, aligning with the temporal resolution of the continuous data. We then add temporal, modality, and session embeddings to the tokens. We train NEDS by masking out tokens according to the masking schemes from (A) and then predicting them with modality-specific decoders. Our multimodal architecture builds on work from other domains (He et al., 2022; Mizrahi et al., 2023; Fang et al., 2024).

we tokenize each 20ms time bin, generating a sequence of $T$ tokens per trial. For the continuous behavioral data, we again utilize temporal tokenization yielding sequences of $T$ tokens. For discrete behavioral data, we replicate each value $T$ times to align with the temporal resolution of the continuous data and convert it into a sequence of $T$ tokens (shown in Figure 1B). To differentiate between modalities, we add learnable modality-specific embeddings to each token (ablated in Appendix G). We also add rotary position embeddings (RoPE) (Su et al., 2024; Azabou et al., 2024) to each token to help the model distinguish data from different timesteps. Tokens from all modalities are concatenated into a single unified sequence, which serves as input to the shared transformer encoder. Tokenizing each modality separately provides flexibility for incorporating additional modalities and cases where modalities are missing.

***Multimodal transformer encoder.*** To process the multimodal sequence of tokens, we employ an encoder-only transformer architecture composed of standard transformer blocks with pre-normalization and feed-forward layers. Central to this architecture is the self-attention mechanism (Vaswani, 2017), which computes pairwise interactions between all tokens in the input sequence. This mechanism enables the model to capture dependencies within modalities, across modalities, and across time steps.

***Modality-specific decoders.*** We use separate linear decoders to reconstruct each modality. This is done by first splitting the transformer encoder output by modality. Then, each modality subset is transformed using its corresponding decoder to reconstruct the original data. This design enables flexible decoding across modalities and allows the model to function effectively even when some modalities are missing, making it suitable for diverse and heterogeneous datasets.

***Session-specific adaptation.*** To enable training across multiple sessions, we use session-specific input matrices to transform neural activity and behavior into fixed-dimensional embeddings (Pandarinath et al., 2018). We also add session embeddings to each token to help the model account for session-specific variations (Azabou et al., 2024; Zhang et al., 2024a). For decoding, we use session- and modality-specific decoders for each animal.

### 3.4. Generative process

To train NEDS, we randomly sample a masking scheme $M_i$ to apply to each sequence $i$ in the batch. We transform neural and behavioral data into a fixed-dimensional space using $W_X^{\text{input}} \in \mathbb{R}^{N \times D}$ and $W_Y^{\text{input}} \in \mathbb{R}^{B \times D}$, where $N$ and $B$ are the dimensions of neural activity and behavior, respectively. We then tokenize the neural data and behav-

ioral data using modality-specific tokenizers (see Section 3.3). After tokenization, we apply the sampled masking scheme by replacing selected tokens with a learned mask token. We then add positional embeddings (PE) to encode three types of structure: modality ($PE_{modality}$), which differentiates neural and behavioral tokens; temporal ordering ($PE_{temporal}$), which captures the sequence position of each token within a session; and session identity ($PE_{session}$), which provides session-specific context. These embeddings are summed with the tokenized inputs before being processed by the transformer, which outputs representations for neural activity ($e_X$) and behavior ($e_Y$). The training is as follows:

$$M_i \sim \mathcal{U}(\text{neural, behavior, within-modal, cross-modal})$$
$$Z_X = \text{Tokenizer}(W_X{}^{\text{input}}(X))$$
$$Z_Y = \text{Tokenizer}(W_Y{}^{\text{input}}(Y))$$
$$Z = M \odot [Z_X, Z_Y]$$
$$Z_{\text{pos}} = Z + PE_{\text{modality}} + PE_{\text{temporal}} + PE_{\text{session}}$$
$$e_X, e_Y = \text{Transformer}(Z_{\text{pos}})$$
$$X \sim \text{Poisson}(W^{\text{rate}}(e_X))$$
$$Y \sim \begin{cases} \text{MSE}(W^{\text{continuous}}(e_Y)) \\ \text{Cross-Entropy}(W^{\text{discrete}}(e_Y)) \end{cases}$$
$$(1)$$

where $\odot$ denotes pointwise multiplication, $W^{\text{rate}} \in \mathbb{R}^{D \times N}$, and $W^{\text{continuous}}$ and $W^{\text{discrete}}$ map the outputs of the transformer to the continuous and discrete behaviors, respectively. We assume that $X$ is modeled by a Poisson emission model with time-varying rates. For continuous behaviors, we assume $Y$ is modeled by a Gaussian with fixed variance, corresponding to minimizing the MSE loss. For discrete behaviors, we assume $Y$ is modeled by a categorical distribution, corresponding to minimizing a cross-entropy loss.

## 4. Evaluation

### 4.1. Tasks

We evaluate the ability of NEDS to model the relationship between neural activity and behavior using two supervised prediction tasks:

1. **Neural encoding**: Predicting spiking activity from the behavioral and task variables, including choice (left/right), "block" prior, wheel speed, and whisker motion energy. For this task, we use all task variables collectively and individually, allowing us to rank each variable's contribution to driving neural responses. We utilize the co-bps metric (Pillow et al., 2008; Pei et al., 2021) which measures neural reconstruction performance using bits per spike.
2. **Neural decoding**: Predicting behavioral and task variables from spiking activity. For choice and block prediction, we use classification accuracy as the evaluation

metric. For decoding wheel speed and motion energy, we quantify performance using single-trial $R^2$, which measures the proportion of variance explained after accounting for the trial average.

### 4.2. Baselines

We compare NEDS against a number of linear and non-linear baselines. For our linear baselines, we compare to a standard linear regression (used in IBL et al. (2023)) and a multi-session reduced-rank neural encoding (Posani et al., 2024) and decoding (Zhang et al., 2024a) algorithm. Both the linear and reduced-rank models use neural activity (or behavior) across all timesteps to predict behavior (or neural activity) at a specific timestep. For our non-linear baselines, we compare NEDS to two recent transformer-based methods: POYO+ (Azabou et al., 2025) and NDT2 (Ye et al., 2024). POYO+ is a multi-session, multi-task neural decoding algorithm based on the original POYO model (Azabou et al., 2024). NDT2 is a multi-session, masked modeling approach for neural self-prediction that can be fine-tuned for decoding. These algorithms represent the current state-of-the-art in neural decoding of spiking activity. Importantly, each model in our paper has a different pretraining objective: POYO+ performs multi-task neural decoding of choice, block, wheel, and motion energy across all animals, NDT2 performs self-supervised prediction of neural activity across all animals, and NEDS performs multimodal, multi-task-masking across all animals (described in Section 3.2). We utilize a re-implemented version of NDT2 for all our analyses.

## 5. Experiments

For all experiments, we evaluate the performance of each model on 10 held-out animals. For these 10 animals, we split the trials into training (70%), validation (10%), and test (20%) sets. The metrics (detailed in Section 4.1) are computed on the test trials for each heldout animal.

### 5.1. Single-session

We first evaluate NEDS on single-session neural encoding and decoding. For this experiment, we train NEDS on the training trials from one heldout animal and then evaluate its performance on the test trials for the same animal. We compare NEDS to the linear baselines and to a unimodal version of NEDS without multimodal, multi-task training (using the same transformer architecture). We conducted extensive hyperparameter tuning by initializing 50 random models with hyperparameters randomly selected from predefined ranges. The model with the best hyperparameters was chosen based on its validation set performance (see Appendix C for the hyperparameter ranges used in this experiment).

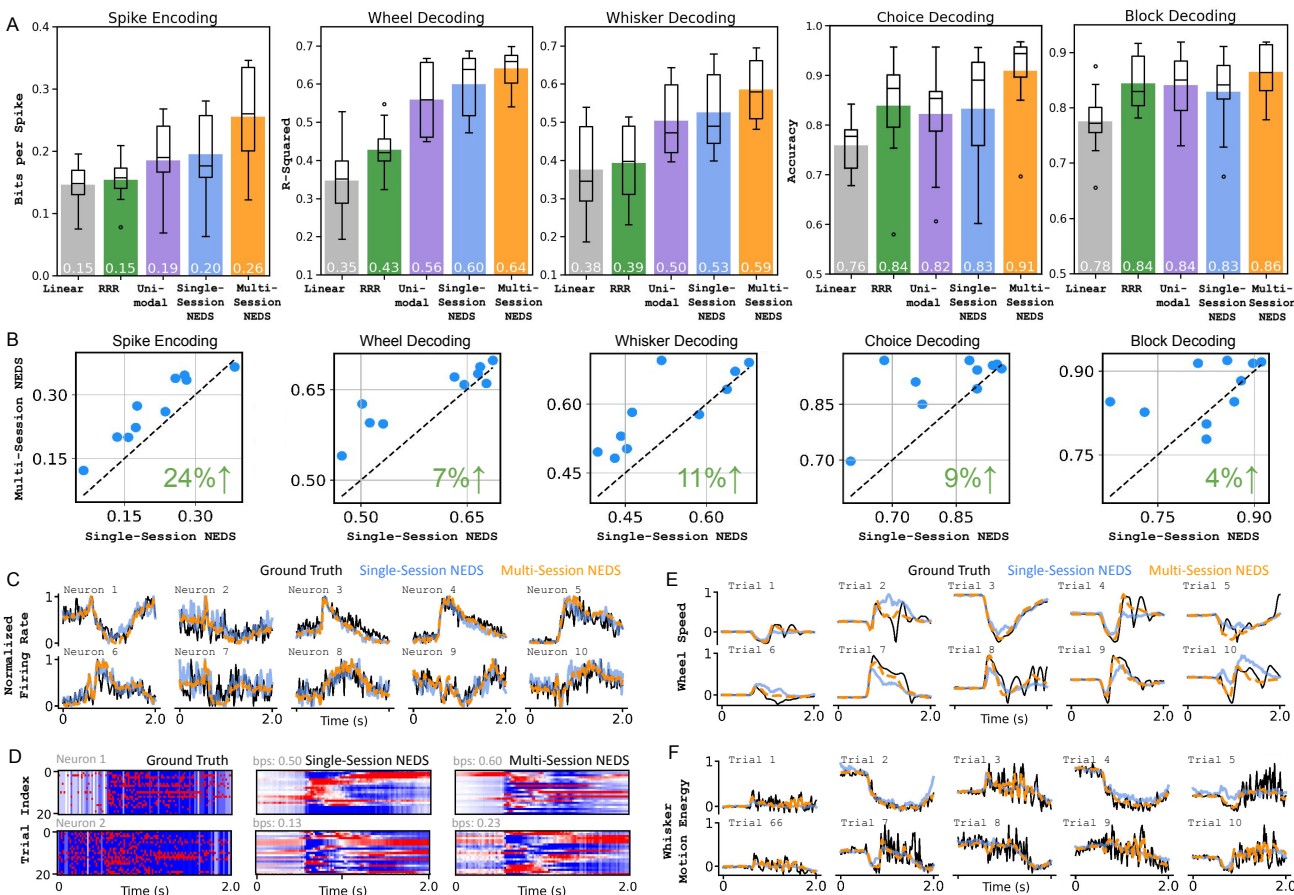

*Figure 2. Quantitative and qualitative evaluation of single-session and multi-session NEDS.* (A) We evaluate multi-session NEDS and single-session NEDS models against our linear baselines and the single-session, unimodal variant of NEDS. Our results show that multi-session NEDS consistently outperforms all baselines across all tasks, while single-session NEDS outperforms all baselines except in block decoding. These findings demonstrate the advantages of multimodal training and cross-animal pretraining for neural encoding and decoding. Among the baseline models, RRR has the fewest parameters (1,000 ~ 20,000 on average). Linear models contain approximately 40,000 to 70,000 parameters on average. Both the single-session unimodal and multimodal NEDS share the same transformer encoder size (~ 3 million parameters). The multi-session NEDS is the largest model with ~ 12 million parameters in its transformer encoder. (B) A scatterplot comparison of multi-session NEDS pretrained on 74 sessions vs. single-session NEDS. Each dot corresponds to an individual session. The green value in the bottom right of each subplot displays the relative improvement of the 74-session NEDS over single-session NEDS. (C) A comparison of the predicted trial-averaged firing rates for single-session and multi-session NEDS against the ground truth trial-averaged spike counts for selected neurons. Predictions from multi-session NEDS more closely matches the ground truth. (D) Each row compares single-session and multi-session NEDS predictions of single-trial variability for a neuron against the ground truth. Single-trial variability is obtained by subtracting the neuron's peristimulus time histogram (PSTH) from its activity in each trial. Only selected trials are shown for visualization purposes. (E, F) The predicted wheel speed and whisker motion energy from both the single-session and multi-session NEDS are shown alongside ground truth behaviors for each trial.

## 5.2. Multi-session

In our multi-session experiments, we evaluate NEDS, POYO+, and NDT2 on the 10 held-out animals after *pretraining* on trials from 73 animals. The goal of this experiment is to assess each model's ability to leverage information from multiple animals to improve generalization to unseen animals. After pretraining, we fine-tune each approach on the training trials from 10 held-out animals, allowing adaptation of session-specific model components

such as session embeddings, input matrices, and linear decoders. We perform multi-task decoding for NEDS and POYO+ and single-task decoding for NDT2 as it does not support multi-task decoding.

Hyperparameter searches become prohibitively expensive in terms of cost and compute when pretraining across multiple animals. To address this, we perform hyperparameter tuning for all methods on a subset of 10 animals. Specifically, we initialize 50 models with hyperparameters randomly sam-

pled from a predefined range and select the best model on the validation performance. Once we identify the optimal architecture, we scale it to 73 animals by increasing model capacity while keeping all other hyperparameters fixed. During fine-tuning, we fix the pretrained model architecture and initialize 30 random models to optimize session-specific hyperparameters, including learning rate, weight decay, and mask ratio (see Appendix C for more details). Our approach attempts to balance practical compute challenges with rigorous fine-tuning (see Section 7 for additional discussion).

### 5.3. Brain region classification with neuron embeddings

Although NEDS is trained to perform neural encoding and decoding, we want to understand what additional information about the neurons is captured by the latent embeddings of our model (e.g., brain region information). There are two matrices in NEDS which store neuron-specific information: the input matrix for neural activity ($W_X^{\text{input}}$) and the output matrix for neural activity ($W^{\text{rate}}$) from Equation 1. For each neuron $k$, we can extract a neuron-specific embedding by taking the corresponding row $k$ in each matrix and concatenating the rows together. We refer to the concatenated embeddings for a neuron $k$ as $C_k$.

We now want to evaluate how well the embedding $C_k$ predicts the brain region where neuron $k$ is located. To test this, we train a linear support vector classifier on the neuron embeddings obtained from the 10 held-out animals after fine-tuning. Our analysis focuses on five main brain regions: the posterior thalamus (PO), lateral posterior nucleus (LP), dentate gyrus (DG), hippocampal CA1 (CA1), and anteromedial visual area (VISa). VISa encompasses several sub-regions which we treat as a single region for this analysis. We evaluate brain region classification using 5-fold cross-validation across all neurons. To understand the importance of multimodal, multi-session training, we compare the neuron embeddings from our pretrained NEDS to single-session unimodal and multimodal versions of NEDS. For the unimodal versions of NEDS, we extract neuron embeddings by taking the corresponding row in $W^{\text{rate}}$ for encoding and $W_X^{\text{input}}$ for decoding (i.e., no concatenation).

## 6. Results

### 6.1. Single-session

In our single-session experiment, we find that NEDS outperforms the linear baselines in neural encoding and decoding of both continuous and discrete behaviors, with the exception of the block prior (shown in Figure 2A). Additionally, multimodal NEDS surpasses its unimodal counterpart on all tasks except block decoding, improving encoding by 5% and decoding by 2–7%, highlighting the advantages of jointly modeling both modalities during training. Most

| All | Choice | Block | Wheel | Whisker |
|---|---|---|---|---|
| $0.27 \pm 0.03$ | $0.06 \pm 0.02$ | $0.10 \pm 0.02$ | $0.12 \pm 0.02$ | $0.11 \pm 0.02$ |

*Table 1. Ranking the importance of task variables for driving neural activity.* We use multi-session NEDS to rank the importance of each variable for neural encoding. We measure encoding performance in bits per spike (bps), where higher values indicate better performance. We show the average and standard error of the encoding performance for each task variable across the 10 animals.

importantly, these results demonstrate that our multi-task-masking strategy enables NEDS to accurately perform both neural encoding and decoding after training.

### 6.2. Multi-session

In our multi-session experiment, we find that NEDS benefits significantly from pretraining on 74 sessions, yielding substantial improvements in both encoding and decoding (Figure 2A) and achieving the best performance across all baselines for all tasks. Figure 2B shows a scatter plot comparing the performance of single-session and multi-session NEDS across each task on the 10 held-out sessions. We find that multi-session NEDS consistently outperforms single-session NEDS across all tasks, with decoding improving by 4–11% and encoding improving by 24%. Figure 2C-F shows the qualitative improvements of NEDS over single-session NEDS for both encoding and decoding.

In Figure 3, we compare NEDS to two state-of-the-art large-scale models for neural decoding: POYO+ and NDT2. After pretraining each approach across 74 sessions and fine-tuning on 10 heldout sessions, we find that NEDS outperforms NDT2 by 11–37% and POYO+ by 1–13% across all decoding tasks. Importantly, NEDS is also able to perform neural encoding unlike these methods.

We also demonstrate that multi-session NEDS can be used to quantify how much each task variable drives neural activity. Table 1 shows a comparison of how well NEDS can predict neural activity from all task variables as well as from each task variable individually. We observe that the movement-related variables, such as wheel speed and whisker motion energy, explain a larger proportion of neural variance than cognition-related variables like choice and block. This is consistent with the findings from IBL et al. (2022). An interesting finding is that encoding performance for all task variables combined is significantly higher than for any individual task variable, suggesting that the task variables capture distinct information about neural activity.

### 6.3. Brain region classification with neuron embeddings

Figure 4 presents an analysis of neuron embeddings extracted from NEDS, following a similar evaluation pipeline used for POYO+ embeddings in Azabou et al. (2025). De-

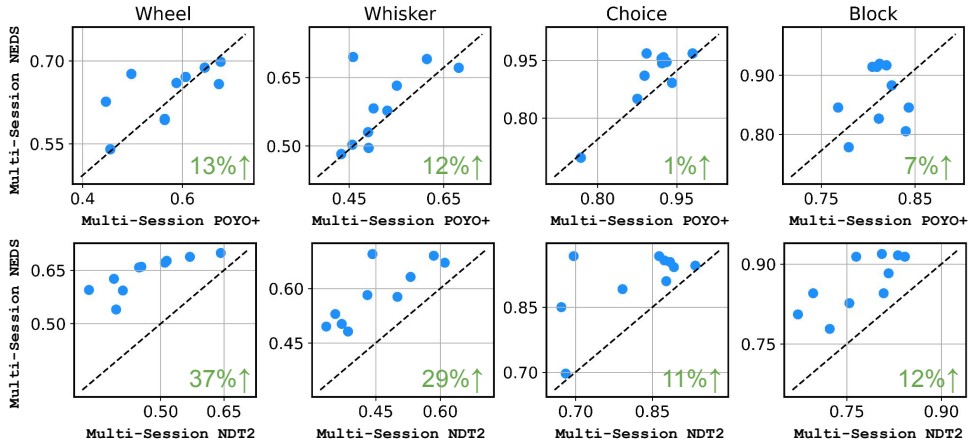

*Figure 3. Comparing NEDS to POYO+ and NDT2.* We compare multi-session NEDS to POYO+ and NDT2 after pretraining on 74 sessions, evaluating all models on neural decoding tasks across 10 held-out sessions. We measure the performance of choice and block decoding with accuracy and the wheel speed and whisker motion energy using single-trial $R^2$. Each dot corresponds to an individual session. The green value in the bottom right of each subplot displays the relative improvement of NEDS over POYO+ and NDT2.

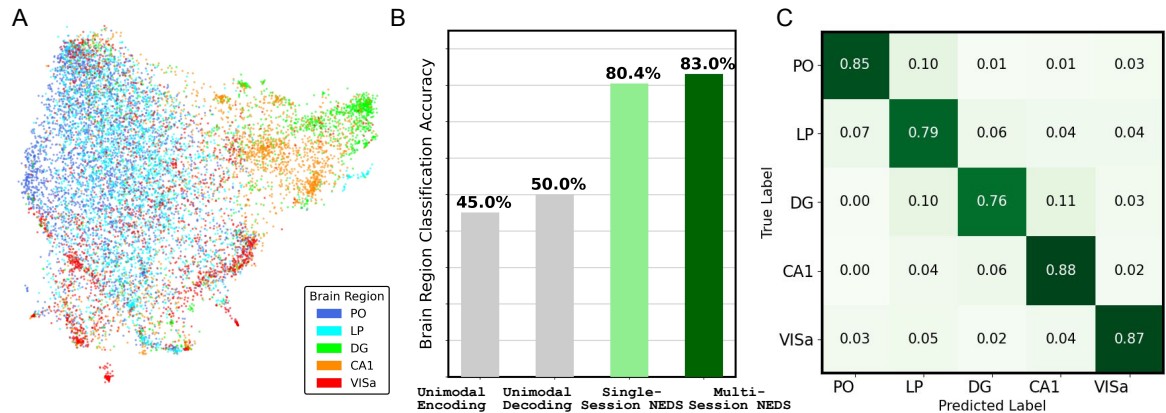

*Figure 4. Brain region classification with neuron embeddings from NEDS.* (A) a UMAP projection of NEDS neuron embeddings (detailed in Section 5.3), color-coded by distinct brain regions. (B) Classification accuracy of brain regions using neuron embeddings obtained from single-session unimodal, multimodal NEDs, and multi-session, mulit-modal NEDS. (C) Confusion matrix showing the brain region classification performance of the neuron embeddings from multi-session NEDS.

spite receiving no explicit brain region information during training, the embeddings of multi-session NEDS exhibit a striking pattern: different brain regions show clear separability (Figure 4A). We compare the brain region classification performance of neuron embeddings extracted from single-session unimodal NEDS, single-session multimodal NEDS, and multi-session NEDS in Figure 4B. We find that the classifier achieves higher accuracy with neuron embeddings from multimodal NEDS than from unimodal NEDS. Pretraining across 74 sessions further improves performance, with multi-session NEDS achieving the highest classification accuracy of all the methods (83%). In Figure 4C, we show the brain region classification confusion matrix for multi-session NEDS. Analyzing the classifier's errors reveals potential similarities between specific brain regions, such as PO and LP. These regions, both located in the pos-

terior thalamus, are known to play key roles in multimodal sensory integration (Allen et al., 2016). Surprisingly, we also find similarities between LP and DG which suggests that these regions might have a subset of neurons with similar functionality. Future work could further investigate the nature of these similarities and their implications for multimodal sensory processing and memory-related functions.

## 7. Discussion

In this work, we introduce an approach for **N**eural **E**ncoding and **D**ecoding at **S**cale (**NEDS**). Our method leverages multi-task-masking, alternating between neural, behavioral, within-modality, and cross-modal masking during training. This approach enables NEDS to accurately predict behavior from neural activity (decoding) and neural activity from behavior (encoding) after training. We demonstrate that pre-

training NEDS across 73 animals improves both encoding and decoding performance on heldout animals, surpassing state-of-the-art methods such as POYO+ and NDT2 on the IBL repeated site dataset. Finally, we demonstrate that the learned neuron embeddings of NEDS display emergent properties: they are highly predictive of the brain regions in each recording without explicit training.

This work has several limitations. First, we rely exclusively on trial-aligned data for pretraining and evaluation of all models. While this approach is standard in systems neuroscience, it limits the amount of data available for pretraining. In contrast, Azabou et al. (2024) pretrained POYO on unaligned data, enabling the use of much larger datasets. Pretraining NEDS with unaligned data could enhance its generalizability and is an exciting direction for future work. Second, due to computational constraints, we were unable to conduct extensive hyperparameter tuning when pretraining models on data from all 73 animals. The development of standardized benchmarks, such as the FALCON benchmark introduced by Karpowicz et al. (2024), will facilitate more rigorous and scalable model comparisons moving forward.

Looking ahead, extending the training paradigm introduced in NEDS to incorporate additional modalities, such as local field potentials and electrophysiological features, presents a promising direction. Recent studies have shown that combining a neuron's activity with its electrophysiological properties enables accurate prediction of its cell-type (Beau et al., 2025; Lee et al., 2024; Yu et al., 2025). Incorporating these additional features into NEDS could enhance its ability to identify and predict cell-type specific functionality. Overall, NEDS introduces a flexible and scalable paradigm for modeling the relationship between neural activity and behavior.

## Acknowledgments

This project was supported by the Wellcome Trust (209558 and 216324), National Institutes of Health (1U19NS123716), the Simons Foundation, the National Science Foundation (NSF award CIF:RI:2212182, NSF CAREER award IIS-2146072), and by DoD OUSD (R&E) under Cooperative Agreement PHY-2229929 (The NSF AI Institute for Artificial and Natural Intelligence), as well as generous gifts from the Alfred Sloan Foundation, the McKnight Foundation, and the CIFAR Azrieli Global Scholars Program.

## Impact Statement

This work advances scalable, multimodal models that improve our understanding of the bidirectional relationship between neural activity and behavior, with potential applications in neuroscience and neurotechnology. While our approach relies on publicly available animal data collected ethically, future extensions to human data will require careful consideration of privacy, consent, and responsible use. Overall, our work contributes to building foundation models that could drive innovations in brain-computer interfaces and neurological therapies.

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

## A. Neural encoding metric (BPS)

We use bits per spike as a metric to evaluate neural encoding performance. The bits per spike for a neuron is defined as follows:

$$\text{bits/spike} = \frac{1}{n_{\text{sp}} \log 2} (\mathcal{L}(r; X_{n,t}) - \mathcal{L}(\bar{r}_{n,:}; X_{n,t})),$$

where $X_{n,t}$ represents the neural activity of the target neuron $n$ at time $t$, $r$ is the inferred firing rate from the model, $\bar{r}_{n,:}$ is the mean firing rate for neuron $n$, and $n_{\text{sp}}$ is its total number of spikes. We then average the bits per spike across all neurons in a test session to get the final performance.

The bits per spike metric is closely related to the deviance, a standard metric generally used in Statistics, e.g., generalized linear models (GLMs) use deviance as a goodness-of-fit measure for a statistical model. The deviance compares the goodness-of-fit of the model of interest, e.g., NEDS, to a baseline null model, where the goodness-of-fit is measured by the model log likelihood. The bps is a normalized version of the deviance metric, which compares the model predictions to the average firing rate of the neuron for the trial. The bps further normalizes the deviance by the spike count so that the metric can be comparable across neurons regardless of whether the neurons are active or inactive.

## B. Masking scheme ablation

In NEDS, we introduce cross-modal and within-modality random masking alongside neural and behavior masking for encoding and decoding (details in Section 3.2). To assess the importance of these additional masking schemes for model performance, we perform an ablation study by training NEDS with each scheme individually removed. We evaluate the models on 10 held-out test sessions for single-session training. The findings are presented in Table 2.

| Masking Scheme Removed | Encoding | Decoding | | | |
|---|---|---|---|---|---|
| | Spike | Choice | Block | Wheel | Whisker |
| None | $0.20 \pm 0.00$ | $0.84 \pm 0.03$ | $0.83 \pm 0.02$ | $0.57 \pm 0.03$ | $0.52 \pm 0.03$ |
| Cross-Modal | $0.20 \pm 0.02$ | $0.83 \pm 0.05$ | $0.83 \pm 0.02$ | $0.50 \pm 0.02$ | $0.46 \pm 0.03$ |
| Within-Modality | $0.10 \pm 0.02$ | $0.84 \pm 0.05$ | $0.81 \pm 0.03$ | $0.52 \pm 0.03$ | $0.46 \pm 0.03$ |

*Table 2. Ablation study of within-modality and cross-modal masking.* We evaluate the impact of the cross-modal and within-modality masking schemes on the performance of NEDS in both single-session experiments. The metrics are computed on the held-out test sessions.

In Table 2, we observe that both within-modality and cross-modal masking have a large impact on the model's performance. Removing the within-modality masking scheme results in a large drop in performance across all metrics, particularly in encoding. Removing the cross-modal masking scheme has a more modest impact on encoding, but has a large effect on continuous behavior decoding.

## C. Model and hyperparxameter details

### NEDS
**Preprocessing.** For both neural encoding and decoding, we utilize the raw spike counts without preprocessing. We normalize the continuous behavior to a range between $-1$ to $1$.

**Losses.** For encoding, we use the Poisson negative log-likelihood as the loss function, while for decoding, we apply cross-entropy loss for discrete behaviors and mean squared error for continuous behaviors.

**Hyperparameters.** To select the optimal model architecture and optimizer configuration for our single-session and multi-session models, we use *Ray Tune* (Liaw et al., 2018) to randomly sample 50 hyperparameter combinations from the ranges specified in Table 3. The hyperparameters for single-session models are chosen by *Ray Tune* based on the ranges specified in Table 3. For each session, we choose 50 randomly sampled models and select the best hyperparameters according to validation performance. To determine the optimal model architecture for multi-session NEDS, we conduct hyperparameter tuning by evaluating 50 randomly sampled models on 10 pretraining sessions. After selecting the architecture, we retain most of its configuration unchanged and only increased the model depth when scaling up to 74 pretraining sessions. When finetuning pretrained models, *Ray Tune* is used only to optimize the mask ratio, weight decay, and learning rate, as the model architecture remains fixed and inherited from the multi-session model. During finetuning, we randomly sample 30

| Hyperparameter | Value Range |
|---|---|
| Embedding Dimension | [128, 256, 512] |
| Head Dimension | [256, 512, 1024] |
| Depth | [5, 6] |
| Mask Ratio | Uniform(0.1, 0.4) |
| Weight Decay | Log-Uniform(0.001, 0.1) |
| Learning Rate | Log-Uniform(0.0001, 0.001) |

*Table 3.* The range of possible NEDS model and optimizer hyperparameters from which *Ray Tune* randomly samples combinations.

| Hyperparameter | Value |
|---|---|
| Embedding Dimension | 256 |
| Head Dimension | 512 |
| Number of Heads | 8 |
| Depth | 22 |
| Mask Ratio | 0.1 |
| Dropout Rate | 0.2 |
| Attention Dropout Rate | 0.4 |
| Weight Decay | 0.01 |
| Learning Rate | 0.003 |
| Batch Size | 1024 |

*Table 4.* Hyperparameters used for training 74-session multimodal NEDS.

combinations of these hyperparameters for each session.

The hyperparameters of our 74-session multimodal model are provided in Table 4. The transformer model comprises 12 million parameters. When including session-specific input matrices and linear decoders, the total number of model parameters increases to 150 million. The dataset consists of roughly 13 million tokens, obtained from 14 hours of neural and behavioral data. The total neuron hours (calculated as the number of neurons multiplied by the number of hours) amounts to roughly 400,000 in total.

*NDT2*

For NDT2, we re-implement the model to align with the workflow of NEDS and ensure consistency in evaluation methods. NDT2 follows an encoder-decoder architecture, specifically a masked autoencoder adapted for neural spike data. The workflow consists of three main steps. As a first step, the model is pretrained with the self-supervised learning (SSL) task of reconstructing masked neural patches. As a second step, the model is fine-tuned for a session of interest with the previous SSL setting. As a final step, the encoder is frozen and a new decoder is trained to decode a task variable. For the SSL pretraining, as in the original NDT2, we use the Poisson negative log-likelihood as the loss function. For decoding, we apply cross-entropy loss for discrete behaviors and mean squared error for continuous behaviors.

For discrete behaviors, we extend NDT2 to classification as the original NDT2 only allows for regression (e.g., velocity decoding). To predict discrete behaviors at the trial level, we avoid generating predictions at every timestamp by using a single query token in the decoder. This query token must be linked to a specific timestamp within the trial to enable temporal attention. We experimented with three options: the first, average, and last timestamp. As expected, using the last timestamp produced the best results, as it allows the query token to leverage the most contextual information due to causal temporal attention masking.

To select the optimal model architecture and optimizer configuration for NDT2 pretraining, we use *Weights & Biases* (Biewald, 2020) to randomly sample 50 hyperparameter combinations from the ranges specified in Table 5. Due to a limited compute budget, we perform our hyperparameter search on 10 randomly selected sessions with an encoder that has 0.5/1.7/6.6 million parameters (depending on the embedding size). The decoder and learning rate scheduler match those of the original NDT2 for pretraining and the other steps. The best hyperparameters for NDT2 pretraining are chosen based on the validation performance (i.e. validation SSL loss).

While scaling up pretraining to 74 sessions, we observed instabilities in the training. To mitigate this, we reduced the

| Hyperparameter | Value Range |
|---|---|
| Embedding | [128, 256, 512] |
| Patch Size | [32, 64] |
| Mask Ratio | Uniform(0.1, 0.5) |
| Weight Decay | Log-Uniform(0.001, 0.1) |
| Learning Rate | Log-Uniform(0.0001, 0.01) |

*Table 5.* The range of possible NDT2 model and optimizer hyperparameters from which *Weights and biases* randomly samples combinations.

learning rate by a factor of 2 (from 0.001 to 0.0005), resulting in more stable training. To give NDT2 more capacity to model 74 sessions of data, we increased the encoder depth from 3 to 6, resulting in a 12.9 million parameters encoder. Additionally, we experiment with scaling the encoder up to 44 million parameters by increasing the encoder depth to 12 and the encoder head dimension to 2,250. However, the pretraining curve shows a similar trend as the 12.9 million parameters encoder, converging at a comparable epoch with similar validation results.

The hyperparameters for NDT2 74-sessions pretraing model are detailed in Table 6.

| Hyperparameter | Value |
|---|---|
| Embedding Dimension | 512 |
| Patch Size | 64 |
| Encoder Depth | 6 |
| Decoder Depth | 2 |
| Encoder Head Dimension | 1024 |
| Decoder Head Dimension | 512 |
| Encoder/Decoder Number of Heads | 4 |
| Encoder/Decoder Dropout | 0.1 |
| Mask Ratio | 0.11 |
| Weight Decay | 0.001 |
| Learning Rate | 0.0005 |
| Batch Size | 1024 |

*Table 6.* Hyperparameters used for pretraining 74-session NDT2.

For the SSL finetuning on individual sessions, we use the best hyperparameter configurations from pretraining. We performed a sweep over the learning rate, but we found it did not result in significant changes in convergence or validation performance. During SSL finetuning, the decoding loss exhibits high variance. To prevent premature stopping and poor checkpoint selection, we apply a running average using a sliding window over the last 10 validation epochs to stabilize loss monitoring.

To obtain the decoding result for a specific session, we freeze the encoder that was trained with SSL finetuning. For each task, we train a new transformer decoder that adapts to the specific task. To obtain the best possible performance, we tested 4 different learning rates (i.e., 0.0004, 0.0002, 0.0001, 0.00005) for each session-task pair. Regardless of the learning rate used, we always report the best session-task result obtained from the best validation result.

### POYO+
Following the tuning strategies for other baselines, we first do a random hyperparameter tuning search using 10 sessions. We perform 100 runs then report the hyperparameters for the best model based on the validation accuracy. We then train a larger model on 74 sessions. The final 74-session model has 3.2 million parameters. The hyperparameters for the model can be found in Table 7. The pretrained model is then finetuned on the 10 held-out sessions. We use early stopping during finetuning and report the test metrics.

### RRR
For the RRR encoder, both the behavior inputs and spike count outputs are z-scored. As a result, the loss function used is mean squared error. During evaluation, the model predictions are converted back to the original data scale. For the RRR decoder, we retain the raw spike count inputs without preprocessing and normalize only the continuous behavior outputs to a range of $-1$ to $1$. We use cross-entropy loss for decoding discrete behaviors and mean squared error for decoding

| Hyperparameter | Value |
|---|---|
| Embedding Dimension | 128 |
| Head Dimension | 64 |
| Number of Latents | 32 |
| Depth | 4 |
| Number of Heads | 8 |
| FFN Dropout | 0.3 |
| Linear Dropout | 0.3 |
| Attention Dropout | 0.3 |
| Weight Decay | 1e-3 |
| Learning Rate | 1e-3 |
| Batch Size | 128 |

*Table 7.* Hyperparameters used for training the 73 session POYO model

continuous behaviors.

To select the optimal model hyperparameter and optimizer configuration for our single-session RRR models, we used *Ray Tune* to randomly sample 50 hyperparameter combinations from the ranges specified in Table 8. The RRR model's maximum allowable rank is 100 (full rank), matching the number of time bins in the IBL data. We select the model rank from a specified range. The RRR encoder and decoder require different optimizers: the encoder uses LBFGS, while the decoder uses AdamW. As a result, they require different ranges of learning rate. For multi-session RRR, we use an encoder rank of 4, a learning rate of 0.001, a weight decay of 0.01, and a batch size of 16.

| Hyperparameter | Value Range |
|---|---|
| Encoder Rank | Randint(2, 50) |
| Encoder Learning Rate | Log-Uniform(0.001, 1) |
| Decoder Rank | Randint(2, 50) |
| Decoder Weight Decay | Log-Uniform(0.001, 0.1) |
| Decoder Learning Rate | Log-Uniform(0.0001, 0.001) |

*Table 8.* The range of possible RRR model and optimizer hyperparameters from which *Ray Tune* randomly samples combinations.

### *Linear*

We begin by preprocessing the input and output of the linear encoder and decoder, standardizing them to follow a normal distribution. Next, we use *scikit-learn*'s (Pedregosa et al., 2011) GridSearchCV function to select the regularization strength from the set [0.0001, 0.001, 0.01, 0.1, 1, 100, 1000, 10000] when fitting the linear encoder and decoder. During evaluation, the model predictions are then converted back to the original data scale.

## D. Model size effect

A model's parameter count provides a general measure of its expressivity, but its ability to generalize also depends on the scale of the training dataset. When excessively large models are trained on limited data, they are more susceptible to overfitting, capturing noise rather than learning meaningful patterns. In practice, models with fewer parameters can outperform larger ones on smaller datasets by maintaining a better balance between model capacity and data constraints. To systematically investigate this trade-off for NEDS, we perform an ablation where we train NEDS with 3 million (3M) and 12 million (12M) parameters on single-session data from the 10 held-out sessions.

In our main experiments, we use a 3-million-parameter model for single-session NEDS and a 12-million-parameter model for 74-session NEDS. As shown in Table 9, model size significantly impacts single-session performance in NEDS. We find that a 12-million-parameter model overfits when trained on a single session, leading to degraded performance in both encoding and decoding tasks. This finding supports our choice of a smaller model for single-session NEDS and a larger model for multi-session training.

| Model Size | Encoding | Decoding | | | |
|---|---|---|---|---|---|
| | Spike | Choice | Block | Wheel | Whisker |
| 3 Million | $0.20 \pm 0.00$ | $0.84 \pm 0.03$ | $0.83 \pm 0.02$ | $0.57 \pm 0.03$ | $0.52 \pm 0.03$ |
| 12 Million | $0.09 \pm 0.06$ | $0.83 \pm 0.15$ | $0.83 \pm 0.06$ | $0.51 \pm 0.07$ | $0.46 \pm 0.10$ |

*Table 9. Effect of model size on NEDS performance.* We evaluate the effect of model size on the performance of NEDS in single-session experiments. All metrics are computed based on the 10 held-out sessions.

## E. Benchmark of NEDS vs. other multi-session models

In Table 10, we show the results of our comparison of NEDS to two state-of-the-art large-scale models for neural decoding: POYO+ and NDT2. We also compare NEDS to a linear multi-session decoder (Zhang et al., 2024a) trained on 74 sessions. We provide the mean and standard error for each method across the 10 held-out sessions.

| Method | Encoding | Decoding | | | |
|---|---|---|---|---|---|
| | Spike | Choice | Block | Wheel | Whisker |
| RRR | - | $0.83 \pm 0.04$ | $0.84 \pm 0.03$ | $0.45 \pm 0.02$ | $0.41 \pm 0.03$ |
| NDT2 | - | $0.81 \pm 0.05$ | $0.79 \pm 0.04$ | $0.47 \pm 0.04$ | $0.46 \pm 0.05$ |
| POYO+ | - | $0.90 \pm 0.02$ | $0.81 \pm 0.01$ | $0.51 \pm 0.03$ | $0.52 \pm 0.02$ |
| NEDS | $0.27 \pm 0.03$ | $0.91 \pm 0.03$ | $0.86 \pm 0.02$ | $0.64 \pm 0.02$ | $0.59 \pm 0.03$ |

*Table 10. Benchmark of NEDS and other multi-session models.* We evaluate all decoding models on choice and block (accuracy), and wheel speed and whisker motion energy (single-trial $R^2$). Each baseline is pretrained on 74 sessions of IBL data and fine-tuned on 10 unseen, held-out sessions. We show the average and standard error for all methods across the 10 sessions.

## F. Training details

*NEDS*

We train both the single-session and multi-session NEDS using the AdamW optimizer. The 74-session models are trained on 16 Nvidia RTX8000 GPUs (each with 48GB memory) in under 2 days for a total of 2000 epochs. Single-session multimodal NEDS and unimodal encoding NEDS can be trained on a single Nvidia A40 GPU in less than 2 hours, also for 2000 epochs. Single-session unimodal decoding NEDS requires less than 30 minutes on one Nvidia A40 GPU for the same number of epochs. Finetuning can be similarly performed using a single GPU. For the extensive hyperparameter tuning experiments, we use 4 Nvidia A40 or V100 GPUs for single-session NEDS (including both training from scratch and fine-tuning), and complete the process in less than 1.5 days.

*NDT2*

We train NDT2 using the AdamW optimizer as original NDT2 does. For the 50 hyper parameters search experiments, we use 1 Nvidia H100 GPU (80GB memory) per experiment. In total, it took 90 hours for running all experiments. For the large scale pretraining experiment on 74 sessions, we use 4 Nvidia H200 GPUs for 4 hours and 30 minutes to reach 600 epochs. For the SSL finetuning experiment on the 10 sessions, we use 1 Nvidia H100 GPU (80GB memory) per session for a total of 2 hours. Finally, for the 160 (= 4 learning rate configurations × 4 tasks × 10 sessions) decoding experiments, we use 1 Nvidia H200 GPU per experiment for a total of 10 hours 30 minutes.

*POYO+*

We use the official implementation of POYO+ which allows us to train the model on four decoding tasks simultaneously. A major difference from the original model is the use of trial-aligned data, with the context window fixed to 2 seconds, and no temporal jittering used during data sampling, which is an important source of data augmentation used during training. We do use unit dropout as data augmentation. When training POYO+, we use the LAMB optimizer (You et al., 2019) with weight decay. The learning rate is held constant, then decayed towards the end of training (last 25% of epochs), using a cosine decay schedule. The model is trained in a supervised manner on four different behavior decoding tasks. To balance the scale of the different tasks, we increase the weight of the regression tasks by a factor of 20. Training is done on a single A100 GPU for 100 epochs, lasting around 6 hours.

*RRR*

We train the single-session RRR encoder using the LBFGS optimizer, which converges quickly under a minute, typically within 20 epochs. For the single-session RRR decoder, we use the AdamW optimizer, requiring less than 20 minutes on a single Nvidia A40 GPU for 1000 epochs. For hyperparameter tuning experiments, we use one CPU for single-session RRR encoder, and one Nvidia A40 GPU for single-session RRR decoder, completing the process within a day. We train the multi-session RRR for 2000 epochs on a single Nvidia A40 GPU, completing the process in two days.

## G. Embedding ablations

We performed an ablation study on single-session data (corresponding to Figure 2) to evaluate the importance of the temporal and modality embeddings. Session embeddings were not ablated, as they are essential for distinguishing data from different sessions. The ablation results, presented in Table 11, demonstrate that the embeddings generally improve performance across all five tasks.

| Model Variant | Encoding (bps) | Choice (Acc) | Block (Acc) | Wheel ($R^2$) | Whisker ($R^2$) |
|---|---|---|---|---|---|
| NEDS (multi-session) | $0.27 \pm 0.08$ | $0.91 \pm 0.08$ | $0.87 \pm 0.05$ | $0.64 \pm 0.05$ | $0.59 \pm 0.08$ |
| NEDS (single-session) | $0.20 \pm 0.06$ | $0.84 \pm 0.12$ | $0.83 \pm 0.07$ | $0.57 \pm 0.09$ | $0.52 \pm 0.10$ |
| NEDS (w/o modality embed) | $0.20 \pm 0.07$ | $0.85 \pm 0.12$ | $0.83 \pm 0.05$ | $0.52 \pm 0.07$ | $0.46 \pm 0.09$ |
| NEDS (w/o temporal embed) | $0.24 \pm 0.07$ | $0.84 \pm 0.12$ | $0.84 \pm 0.05$ | $0.52 \pm 0.08$ | $0.48 \pm 0.09$ |
| NEDS (w/o temporal + modality) | $0.25 \pm 0.07$ | $0.86 \pm 0.11$ | $0.83 \pm 0.06$ | $0.53 \pm 0.07$ | $0.47 \pm 0.10$ |

*Table 11. Ablation of modality and temporal embeddings in* NEDS. We ablate the embeddings for single-session NEDS and test the performance on the IBL repeated site dataset. Reported values are means $\pm$ standard deviations across 10 test sessions.

## H. NEDS on a monkey reaching dataset

We tested NEDS on the MC-RTT primate motor task dataset (Pei et al., 2021), which differs significantly from the IBL visual decision-making dataset in several ways: (1) MC-RTT uses Utah arrays, whereas IBL relies on Neuropixels recordings, (2) MC-RTT involves monkeys, while IBL uses mice. (3) MC-RTT focuses on a random target reaching motor task, unlike the visual decision-making task in IBL (4) MC-RTT data is unaligned, compared to the trial-aligned structure in IBL.

As the MC-RTT dataset contains only a single recording session, we compared the single-session variant of NEDS against an MLP decoder. Encoding quality was measured via bits-per-spike (bps), and decoding performance for finger velocity was measured using $R^2$. We utilized the MLP and data splits from `https://github.com/seanmperkins/bci-decoders/` which has implementations of a few common neural decoding algorithms. The neural and behavioral data was binned at 20ms. To predict each time step, we used a time history of 13 bins which is the default in the repository. We tuned the learning rate of the MLP, keeping other parameters (e.g. model sizes and optimizers) fixed to the defaults provided. We trained 30 random models for NEDS and chose the best validation performance, similar to the procedure in our main text. The results, summarized in Table 12, show that NEDS can work well across recording modalities, species, and tasks.

| Method | Encoding (bps) | Decoding (Vel $R^2$) |
|---|---|---|
| MLP | - | 0.66 |
| Unimodal NEDS | 0.04 | 0.65 |
| Multimodal NEDS | 0.07 | 0.72 |

*Table 12. Benchmark of NEDS on the MC-RTT dataset.* We evaluate single-session NEDS and a baseline MLP on finger velocity decoding (single-trial $R^2$). Multimodal NEDS outperforms both the unimodal variant and MLP.

