# OpenReview forum: "Neural Encoding and Decoding at Scale"
_ICML.cc/2025/Conference — ICML 2025 spotlightposter_

### Official Review · Reviewer_Fbeh · 2025-02-24

**Overall Recommendation:** 3

**Summary:**

This article introduces a multimodal, multi-task model named "Neural Encoding and Decoding at Scale (NEDS)" for large-scale neural encoding and decoding. The model employs a novel multi-task masking strategy, enabling simultaneous bidirectional prediction between neural activity and behavior—predicting neural activity from behavior (encoding) and predicting behavior from neural activity (decoding). NEDS was pre-trained on a large-scale multi-animal dataset and fine-tuned on new animal data, demonstrating exceptional performance and generalization capabilities.

**Claims And Evidence:**

Convincing enough.

**Essential References Not Discussed:**

Li Y, Lou A, Xu Z, et al. NeuroBOLT: Resting-state EEG-to-fMRI Synthesis with Multi-dimensional Feature Mapping[J]. Advances in Neural Information Processing Systems, 2025, 37: 23378-23405.

**Experimental Designs Or Analyses:**

The paper only compares two methods, POYO+ and NDT2, on a single dataset. Although some performance improvements were achieved, the overall framework does not differ fundamentally from prior work. The experimental section is also limited to evaluating performance in terms of encoding and decoding, lacking insightful scientific findings.

**Methods And Evaluation Criteria:**

The methodology presented in this paper bears resemblance to previous works such as POYO+ and NDT2. Moreover, the IBL repeated site dataset serves as a benchmark within the domain, underscoring the significance of this study in advancing the alignment research between neural and behavioral modalities. The contributions of this paper are thus noteworthy for their potential to enrich our understanding of the intricate interplay between neural activity and behavior.

**Other Comments Or Suggestions:**

No.

**Other Strengths And Weaknesses:**

My main concern lies in the potential value of the contribution of the neuro-behavioral conversion presented in this article. Compared to previous works, this article seems more like an incremental project and does not address fundamental issues in the field, such as whole-brain alignment for neuro-behavioral conversion or conversion specific to the human brain, among others.

**Questions For Authors:**

I don't have any more questions; I am looking forward to the author's response to the concerns I raised earlier.

**Relation To Broader Scientific Literature:**

NDT2 is a Transformer-based spatiotemporal encoder-decoder model that undergoes unsupervised pre-training through Masked Autoencoding (MAE). And POYO+ is a novel multi-task, multi-session neural decoding approach capable of decoding neural activity from different cell types and brain regions.

This article combines encoding and decoding to form a neuro-behavioral conversion, an idea which I find unsurprising given its similarity to concepts already prevalent in the paper by Li et al. [1].

[1] Li Y, Lou A, Xu Z, et al. NeuroBOLT: Resting-state EEG-to-fMRI Synthesis with Multi-dimensional Feature Mapping[J]. Advances in Neural Information Processing Systems, 2025, 37: 23378-23405.

**Theoretical Claims:**

The use of mask-based pre-training for Transformers is quite common. I am curious whether the three embeddings—Modality Embedding, Temporal Embedding, and Session Embedding—are truly effective. Can the author provide evidence to support this?

---

> ### Author Rebuttal · Authors · 2025-04-01
>
> > The use of mask-based pre-training for Transformers is quite common. I am curious whether the three embeddings—Modality Embedding, Temporal Embedding, and Session Embedding—are truly effective...
>
> While masked modeling is a common objective for training transformers, it is not yet clear which masking schemes work well for modeling neural activity and behavior. We aimed to address this question in our paper and with our proposed method.
>
> We appreciate the suggestion to ablate the different embeddings for NEDS. To do this, we performed an ablation study on single-session data (10 test sessions) to evaluate the importance of the temporal and modality embeddings. Session embeddings were not ablated, as they are essential for distinguishing data from different sessions. The ablation results, presented in the following table, demonstrate that the embeddings generally improve performance across the 5 tasks we evaluate (with a few exceptions). We want to note that these are single-session ablations and will be more sensitive to hyperparameters.
>
> |                          | Encoding        |      | Decoding                         |      |      |      |
> |--------------------------|----------------|------|----------------------------------|------|------|------|
> |                          | Encoding (bps) | Choice (Acc) | Block (Acc) | Wheel (R2) | Whisker (R2) |
> | NEDS (multi-session)     | **0.267** ±0.080 | **0.909** ±0.084 | **0.865** ±0.052 | **0.641** ±0.051 | **0.586** ±0.083 |
> | NEDS (single-session)    | 0.203 ±0.062 | 0.840 ±0.115 | 0.827 ±0.072 | 0.568 ±0.089 | 0.523 ±0.097 |
> | NEDS (w/o modality embed) | 0.198 ±0.070 | 0.846 ±0.116 | 0.831 ±0.046 | 0.516 ±0.066 | 0.459 ±0.086 |
> | NEDS (w/o temporal embed)    | 0.243 ±0.070 | 0.842 ±0.121 | 0.835 ±0.053 | 0.523 ±0.082 | 0.478 ±0.085 |
> | NEDS (w/o temporal + modality embed) | 0.246 ±0.069 | 0.859 ±0.106 | 0.830 ±0.063 | 0.530 ±0.067 | 0.469 ±0.095 |
>
> > The paper only compares two methods, POYO+ and NDT2, on a single dataset. Although some performance improvements were achieved, the overall framework does not differ fundamentally from prior work. The experimental section is also limited to evaluating performance in terms of encoding and decoding, lacking insightful scientific findings.
>
> Prior work in modeling spiking neural activity and behavior focuses on modeling a single direction of the relationship (i.e., decoding or encoding). NEDS introduces a new framework for modeling neural activity and behavior by jointly modeling the two modalities, allowing for simultaneous encoding and decoding at test time (POYO+ and NDT2 solely focus on decoding). We believe this multimodal approach is novel for this domain.
>
> We agree that more experiments are needed to extract scientific insights. However, encoding and decoding remain key tools for understanding what and how information is represented in the brain. For example, the International Brain-wide Map [1] used these methods to characterize how visual, sensory, and motor information are distributed across the mouse brain. We are excited to extend this with NEDS, leveraging large-scale, multi-animal data to capture richer information and reveal how neuronal functional profiles relate to brain anatomy, as shown in Figure 4.
>
> > My main concern lies in the potential value of the contribution of the neuro-behavioral conversion presented in this article. Compared to previous works, this article seems more like an incremental project and does not address fundamental issues in the field, such as whole-brain alignment for neuro-behavioral conversion or conversion specific to the human brain...
>
> We thank the author for the suggested citation which we would be happy to include. We want to emphasize that we strongly believe that our contribution is not incremental. Currently, there are very few foundation models trained on spiking data, contributing to the novelty of our approach. Also, current foundation modeling approaches for spiking activity have primarily focused on decoding, whereas, to the best of our knowledge, no prior work has unified neural encoding and decoding.
>
> We note key differences from NeuroBOLT [1]: NeuroBOLT translates between neural modalities (fMRI and EEG) without modeling behavior, and uses separate encoders per modality, unlike NEDS, which uses a shared transformer to unify representations.
>
> We also feel that the critique related to whole-brain alignment does not apply to spiking recordings which are spatially restricted. Whole-brain alignment is a more tractable problem in fMRI research due to its broader spatial coverage, but is not typically addressed in invasive electrophysiology.
>
> [1] International Brain Laboratory, et al. "A Brain-Wide Map of Neural Activity during Complex Behaviour." bioRxiv (2024).
>
> [2] Li Y, Lou A, Xu Z, et al. NeuroBOLT: Resting-state EEG-to-fMRI Synthesis with Multi-dimensional Feature Mapping[J]. Advances in Neural Information Processing Systems, 2025, 37: 23378-23405.

---

### Official Review · Reviewer_MJ1y · 2025-03-12

**Overall Recommendation:** 4

**Summary:**

This paper introduces Neural Encoding and Decoding at Scale (NEDS), a multimodal, multi-task model that simultaneously performs neural encoding (predicting neural activity from behavior) and neural decoding (predicting behavior from neural activity) by bridging behaviors and neural activity with a shared masked training Transformer. The framework is evaluated on 83 mice and shows good results in single-session and multi-session cases.

## update after rebuttal
Thanks for the responses. I keep my original score and hope to see these supplements and further discussion in the camera-ready version.

**Claims And Evidence:**

Yes.

**Essential References Not Discussed:**

No. The reference is well.

**Experimental Designs Or Analyses:**

Yes, good.

**Methods And Evaluation Criteria:**

Yes.

**Other Comments Or Suggestions:**

1. It could be better to extend the results to other recordings, or you may discuss the heterogeneity of the neuropixel data to show the capability of the model.
2. Just one hold-out test is a bit thin, and it would be beneficial to do at least one or two more validations to make sure the results are stable, albeit at a cost.
3. Have to test the impact of the token length? How do you prepare the continuous and discrete data to cover an overall behavior?
4. Is there any difference between the data from different labs (total 10, mentioned in the section 3.1)?
5. It would be beneficial to give some details about the four kinds of behaviors, or give some references.
6. Have you introduced the design of the objective functions and some model details, such as the position embeddings.
7. Is it possible to test the performance in the animal-independent condition, which means test the performance with finetuning on the test 10 mice?
8. Statistical analysis would make the results better.

**Other Strengths And Weaknesses:**

Strengths: That’s a valuable research focusing on handling neural encoding and decoding with a shared model. Substantial experiments show effectiveness of multi-modal training to achieve good results.

Weaknesses: 1) The evaluation was performed with fixed 10 mice held out from 83 mice, which may introduce randomness. 2) In my view, the work successfully validates the effectiveness of using multi-session data for pretraining and training shared models with brain and behavior data, instead of “scaling” for a large model. Therefore, the title may not be suitable for the content, though that’s cool with a short title.

**Questions For Authors:**

Please see the suggestions part.

**Relation To Broader Scientific Literature:**

Yes, some recent works have been involved for comparison.

**Theoretical Claims:**

Yes, that's an empirical but interesting study.

---

> ### Author Rebuttal · Authors · 2025-04-01
>
> > It could be better to extend the results to other recordings, or you may discuss the heterogeneity of the neuropixel data to show the capability of the model.
>
> We agree with the reviewer's suggestion to extend NEDS to other datasets. To address this, we are currently training NEDS on a primate motor task dataset to show its generalizability across different recording setups, species, and behavioral tasks. In future work, we are excited to extend our approach to larger and more diverse datasets.
>
> > Just one hold-out test is a bit thin, and it would be beneficial to do at least one or two more validations to make sure the results are stable, albeit at a cost.
>
> We agree that additional validation would strengthen the evaluation. Our results rely on a fixed hold-out set due to computational and experimental constraints, and while more rigorous cross-validation would provide a more accurate assessment of model differences, it is currently challenging given the scale of our datasets and models. The development of shared benchmarks will help enable more robust comparisons across different approaches; we plan to participate in such benchmarks in future work.
>
> > Have to test the impact of the token length? How do you prepare the continuous and discrete data to cover an overall behavior?
>
> This is a great question and something we have not explored. We utilize the tokenization scheme from [1] for our model where the neural data is binned at a specific resolution (20ms) and then each time step is passed into the transformer as a token. We fix the context length to 2 seconds, aligned to movement onset, for all trials and animals. The model performance is more likely influenced by the time bin size rather than token length, as suggested in previous work [1]. We are currently training NEDS on two different bin sizes on a monkey dataset. We are excited to further explore this in future work. For more details on how we tokenized continuous and discrete behaviors, please refer to Section 3.3, “Modality-Specific Tokenization,” in the main paper.  At a high-level, the discrete data is transformed into tokens that are repeated multiple times to match the resolution of the continuous data.
>
> > Is there any difference between the data from different labs (total 10, mentioned in the section 3.1)?
>
> This is an interesting question! Part of the goal of the International Brain Laboratory (IBL) was to reproduce the same experiment and data across multiple labs. To do this, the IBL utilized a standardized experimental protocol and quality metrics to ensure consistent data quality across labs [2]. As can be seen in Figure 7 of [2], the authors conducted an experiment using all neurons in the IBL repeated site datasets to predict their brain region and lab identity. The findings indicate that while brain region identity can be reliably decoded from single-neuron profiles, lab identity cannot be inferred from the data.
>
> > It would be beneficial to give some details about the four kinds of behaviors, or give some references.
>
> We appreciate the feedback and will incorporate more descriptions of the behaviors into the final camera-ready.
>
> > Have you introduced the design of the objective functions and some model details, such as the position embeddings.
>
> Yes, details about the objective function and position embeddings can be found in Section 3.3, “Architecture,” of the main paper.
>
> > Is it possible to test the performance in the animal-independent condition, which means test the performance with fine-tuning on the test 10 mice?
>
> Currently, fine-tuning is required for evaluating these models to align the session and neuron-specific weights into a shared representation. Zero-shot performance of these models is an exciting future direction.
>
> > Statistical analysis would make the results better.
>
> We agree that statistical analysis would improve our results. To address this in the limited time window during the rebuttal, we are currently re-running our single-session analysis with different random seeds to determine whether model performance was significantly influenced by randomness. We plan to include these results in our response once the experiments are complete.
>
> [1] Pei, Felix, et al. "Neural Latents Benchmark'21: Evaluating latent variable models of neural population activity." arXiv preprint arXiv:2109.04463 (2021).
>
> [2] International Brain Laboratory, et al. "Reproducibility of in vivo electrophysiological measurements in mice." bioRxiv (2024): 2024-12.

---

> > ### Comment · Reviewer_MJ1y · 2025-04-07
> >
> > The authors haven't directly addressed most of my concerns but attributed them to future work and the camera-ready version.
> >
> > Especially:
> > 1. The limitation of validation (using 10 mice held out from 83 mice).
> > 2. The choice of key parameters, such as the token length.
> > 3. Data heterogeneity across labs.
> > 4. Statistical analysis.

---

> > > ### Author Response · Authors · 2025-04-08
> > >
> > > We would like to thank all the reviewers for their patience. **We are unable to post the updated results for all the reviewers** so we hope that they can look at this response.
> > >
> > > **Reviewer MUVE, geJL and MJ1y**:
> > >
> > > We thank the reviewer for suggesting an evaluation of our model’s **generalizability across diverse datasets, tasks, and unaligned data**. To this end, we tested NEDS on the MC-RTT primate motor task dataset [1], which differs significantly from the IBL visual decision-making dataset in several ways: (1) MC-RTT uses Utah arrays, whereas IBL relies on Neuropixels recordings; (2) MC-RTT involves monkeys, while IBL uses mice; (3) MC-RTT focuses on a random target reaching motor task, unlike the visual decision-making task in IBL; and (4) MC-RTT data is unaligned, compared to the trial-aligned structure in IBL. As the MC-RTT dataset contains only a single recording session, we compared the single-session variant of NEDS against a state-of-the-art MLP decoder. Encoding quality was measured via bits-per-spike (bps), and decoding performance for finger velocity was measured using R2.
> > >
> > > We utilized the MLP and data splits from https://github.com/seanmperkins/bci-decoders/. We tuned the learning rate of the MLP, keeping other parameters fixed to the defaults provided in the repository. We trained 30 random models for NEDS and chose the best validation performance similar to what we did in our main text. The results, summarized in the table below, show that NEDS works well across recording modalities, species, and tasks.
> > >
> > > | Method              | MC-RTT (20 ms) | Encoding (bps) | Decoding (Vel R2) |
> > > |---------------------|----------------|----------------|--------------------|
> > > | MLP                 | NA             |                | 0.66440            |
> > > | Unimodal NEDS       | 0.03711        |                | 0.65029            |
> > > | Multimodal NEDS     | **0.07168**        |                | **0.71786**            |
> > >
> > > [1] Pei, Felix, et al. "Neural Latents Benchmark'21: Evaluating latent variable models of neural population activity." arXiv preprint arXiv:2109.04463 (2021).
> > >
> > > ----------------------------
> > >
> > > **Reviewer MJ1y** (statistical analysis):
> > >
> > > We re-ran our single-session analysis with different random seeds to determine whether model performance was significantly influenced by randomness. The results, shown in the table below, indicate that performance is consistent across seeds for all tasks—except for whisker motion energy decoding, which exhibits some variability. We expect that multi-session pre-training would reduce this variability as it is easy to overfit during single-session training.
> > >
> > > | Seed     | Encoding (bps)       | Choice (Acc)        | Block (Acc)         | Wheel (R2)             | Whisker (R2)           |
> > > |----------|----------------------|----------------------|----------------------|--------------------|--------------------|
> > > | seed-42  | 0.202872 ± 0.06201   | 0.83986 ± 0.11547    | 0.82679 ± 0.07201    | 0.56819 ± 0.08910  | 0.52340 ± 0.09704  |
> > > | seed-43  | 0.20549 ± 0.07061    | 0.82951 ± 0.11135    | 0.82841 ± 0.07333    | 0.52912 ± 0.07930  | 0.45787 ± 0.10177  |
> > > | seed-44  | 0.18999 ± 0.06888    | 0.83449 ± 0.13023    | 0.82645 ± 0.05390    | 0.52050 ± 0.06860  | 0.47325 ± 0.09959  |
> > > | Average | 0.19945 ± 0.00830 | 0.83462 ± 0.00518 | 0.82722 ± 0.00105 | 0.53926 ± 0.02542 | 0.48484 ± 0.03427 |

---

### Official Review · Reviewer_geJL · 2025-03-13

**Overall Recommendation:** 5

**Summary:**

This paper proposes NEDS, a multimodal, multi-task auto-encoder to learn meaningful representations of neural activity and behavior. In brief, the model is based on an encoder-only transformer that tokenized spikes in a similar scheme to NDT (linear projection of binned spikes), as well as both continuous discrete behaviors, then decodes the same information from a multi-head decoder. This configuration allows the authors to perform multiple tasks: predicting behavior from neural activity, neural activity from behavior, and solving different masking problems. The authors demonstrate their model on the International Brain Lab (IBL) dataset, showcasing that it performs better at many of tasks than a slate of baseline models: linear decoders, reduced rank regression, unimodal models, POYO+ and NDT2. They find that the latent embeddings from the model correlate with which brain regreion the data came from, bringing confidence that it has a learned a meaningful representation of the neural data.

**Claims And Evidence:**

Yes, the claims are well-substantiated and the evidence is clear and compelling.

**Essential References Not Discussed:**

None.

**Experimental Designs Or Analyses:**

I did check the soundness of their experiments and analyses. Again, this is pretty well-trodden territory, the IBL dataset is a well-understood dataset, used prior in many papers, including in prior work from Zhang et al. (2024), and the dataset selection, model selection, etc. seem appropriate and in line with the literature.

**Methods And Evaluation Criteria:**

To evaluate this model, the key question is whether the proposed method works better than prior methods: they show this through extensive and convincing evaluation with an appropriate slate of benchmark models. Indeed, the extent of the improvements they demonstrate is remarkable compared to prior art; I was impressed with how much better this did than their well-tuned baseline. A secondary consideration is whether the model gives new insights, and their Figure 4 is an interesting proof of concept of that; though I should add that region decoding is a fairly artificial and rather easy task, it is nevertheless reassuring that this works.

**Other Comments Or Suggestions:**

The reference to Azabou et al. (the original POYO paper) has the wrong date, it was published at NeurIPS 2023, not 2024.

**Other Strengths And Weaknesses:**

There are nice figures, the text is well-written. It would be nice to see this applied to other datasets in a follow-up paper, but there's plenty here to warrant publication.

**Questions For Authors:**

The paper is very clear, no further questions.

**Relation To Broader Scientific Literature:**

The proposed method builds on prior work, including some of the insights from POYO (a multi-head decoder), NDT (tokenization scheme), and the universal translator from Zhang et al. 2024 (multi-mask scheme). The key innovation is using one model to co-embed behavior and spikes. This deserves a publication in and of itself.

**Theoretical Claims:**

No theoretical claims in this paper.

---

> ### Author Rebuttal · Authors · 2025-04-01
>
> We appreciate that the reviewer found our paper well-written and our evaluation convincing. We agree that evaluating NEDS on additional datasets would make our paper stronger. In response, we are currently applying NEDS to a primate motor task dataset (MC-RTT [1]) to demonstrate its generalizability across different recording modalities, species, behavioral tasks, and data structures. We plan to include these results in our response once the experiments are complete.
>
> [1] Pei, Felix, et al. "Neural Latents Benchmark'21: Evaluating latent variable models of neural population activity." arXiv preprint arXiv:2109.04463 (2021).

---

### Official Review · Reviewer_MVUE · 2025-03-17

**Overall Recommendation:** 4

**Summary:**

The paper introduces Neural Encoding and Decoding at Scale (NEDS), a multimodal, multi-task model designed to simultaneously predict neural activity from behavior (encoding) and behavior from neural activity (decoding) using large-scale, multi-animal datasets. NEDS employs a novel multitask-masking strategy that alternates between neural, behavioral, within-modality, and cross-modality masking, implemented within a transformer-based architecture. The model is pretrained on the International Brain Laboratory (IBL) repeated site dataset, comprising Neuropixels recordings from 83 mice performing a visual decision-making task, and fine-tuned on held-out animals. The main findings include: (1) NEDS achieves state-of-the-art performance in both encoding and decoding compared to baselines like POYO+ and NDT2; (2) performance scales with pretraining data and model capacity; and (3) NEDS’s latent embeddings exhibit emergent properties, predicting brain regions with 83% accuracy without explicit training

**Claims And Evidence:**

The claims made in the submission are largely supported by clear and convincing evidence. The authors claim that NEDS outperforms existing large-scale models in both encoding and decoding, which is substantiated by quantitative comparisons with POYO+, NDT2, and linear baselines across tasks like choice, block prior, wheel speed, and whisker motion energy. The evidence includes performance metrics such as bits per spike (bps) for encoding and accuracy/$R^2$ for decoding, computed on 10 held-out animals. The claim of scalability with pretraining data is supported by the improved performance of multi-session NEDS over single-session NEDS. The emergent property of brain region prediction is convincingly demonstrated through a linear classifier achieving 83% accuracy on neuron embeddings. No claims appear problematic, as the results are consistently backed by experimental data and visualizations.

**Essential References Not Discussed:**

N/A

**Experimental Designs Or Analyses:**

I reviewed the experimental designs and analyses in Sections 5 and 6, focusing on single-session, multi-session, and brain region classification experiments. The designs are sound: pretraining on 74 sessions and fine-tuning on 10 held-out sessions is a standard approach, and the train-validation-test split (70%-10%-20%) is appropriate. The ablation study on masking schemes (Appendix B) validly isolates the contribution of each component, showing that within-modality and cross-modal masking enhance performance. Hyperparameter tuning using Ray Tune (Appendix C) is rigorous. The brain region classification experiment is well-executed, using 5-fold cross-validation and comparing unimodal vs. multimodal embeddings.

**Methods And Evaluation Criteria:**

The proposed methods and evaluation criteria are well-suited to the problem of modeling bidirectional relationships between neural activity and behavior. The multitask-masking strategy is a logical extension of masked modeling techniques, allowing flexibility for both encoding and decoding within a single framework. The use of the IBL dataset, with its standardized recordings across 83 mice, is an appropriate benchmark for evaluating scalability and generalization in systems neuroscience. Evaluation metrics—bits per spike for encoding and accuracy/$R^2$ for decoding—are standard and meaningful for assessing neural prediction tasks. The comparison with state-of-the-art models (POYO+, NDT2) and linear baselines ensures a robust evaluation. However, the reliance on trial-aligned data and simple behavioral variables (e.g., wheel speed, choice) may limit the generalizability to more complex tasks, though this is acknowledged as a limitation in Section 7.

**Other Comments Or Suggestions:**

N/A

**Other Strengths And Weaknesses:**

Strengths: The paper’s originality lies in its creative combination of masked modeling and multi-task learning, enabling a flexible, bidirectional model—a significant advance over task-specific predecessors. The clarity of explanation is strong. The potential for real-world impact, such as brain-computer interfaces, enhances its significance. The emergent brain region prediction also adds an valuable insight.

Weaknesses: The reliance on mice data and simple behavioral tasks (e.g., wheel speed, whisker motion) limits the demonstration of NEDS’s capability for complex behaviors like visual decoding. While the authors suggest extensions to other modalities (Section 7), the current scope feels narrow given the “foundation model” ambition. The computational constraint on hyperparameter tuning for multi-session models (Section 7) is a minor weakness, though mitigated by practical tuning on a subset.

**Questions For Authors:**

Generalizability to Complex Tasks: The evaluation focuses on simple behaviors (e.g., choice, wheel speed) in mice. Have you tested or plan to test NEDS on more complex tasks, such as visual decoding or multi-step decision-making?

Potential for unaligned data: The authors mentioned this in the limitations. Given POYO is capable of training on unaligned data and the claim that NEDS's potential for this extension, how would this be possible for NEDS and why isn't it explored in this work?

**Relation To Broader Scientific Literature:**

NEDS builds on prior work in neural encoding and decoding, it's main algorithmic inspiration being He's 2022 work on Masked Autoencoders for computer vision tasks. Based on this idea, it extends unimodal approaches like POYO+ (decoding-focused) and NDT2 (decoding-capable) by unifying encoding and decoding, addressing a gap noted in the literature.

**Theoretical Claims:**

The paper does not present formal theoretical proofs requiring verification.

---

> ### Author Rebuttal · Authors · 2025-04-01
>
> We appreciate the reviewer’s suggestion to evaluate the generalizability of our model on additional datasets, tasks, and unaligned data. Given the complexity of the neural recordings we analyze in our paper, spanning multiple brain regions and animals, we intentionally focused on a small set of well-defined, trial-aligned behaviors (e.g., choice) for our evaluation. We agree with the reviewer that it would be interesting to test NEDS across more complex datasets and tasks. To address this, we are currently training multiple baselines models and NEDS on the MC-RTT primate motor task dataset [1], which differs significantly from the IBL visual decision-making task and is also unaligned. We plan to include these results in our response once the experiments are complete.
>
> [1] Pei, Felix, et al. "Neural Latents Benchmark'21: Evaluating latent variable models of neural population activity." arXiv preprint arXiv:2109.04463 (2021).

---

### Decision · Program_Chairs · 2025-05-01

**Decision:**

Accept (spotlight poster)

**Comment:**

The paper is along the recent trend of foundation models in neuroscience. It establishes a multi-modal multi-task encoding-decoding approach to predict neural activity from behavior and vice versa. The paper tests this approach in a single-session and multi-session setting, establishing that pre-training on multiple sessions improves performance over the single-session case and prior approaches like NDT2 and Poyo+. All reviewers applaud the paper as being well written and making a noteworthy contribution (albeit with varying degree of enthusiasm). Hence I believe this is a strong case for acceptance.

A few comments to the authors for the camera-ready version:
- The ablation table in response to reviewer Fbeh seems misleading. If I read your rebuttal correctly, these experiments were done on single-session NEDS but you include and boldface multi-session NEDS in this table, suggesting somewhat misleadingly that removing the modality and temporal embeddings decreases performance, while it does not when properly compared to the single-session version in row 2 (in fact, performance increases in one case and is unchanged in another).
- That multi-animal pretraining improves performance has been shown several years ago in the context of visual encoding models by [Lurz et al., ICLR 2021](https://openreview.net/forum?id=Tp7kI90Htd) – would probably make sense to mention it in the related work.